# Ising Ladder with Four-Spin Plaquette Interaction in a Transverse Magnetic Field

**DOI:** 10.3390/e25121665

**Published:** 2023-12-16

**Authors:** Maria Eugenia S. Nunes, Francisco Welington S. Lima, Joao A. Plascak

**Affiliations:** 1Departamento de Física, Universidade Federal de Ouro Preto, Campus Universitário Morro do Cruzeiro, ICEB, Ouro Preto 35400-000, MG, Brazil; mariaeugenia@ufop.edu.br; 2Dietrich Stauffer Computational Physics Lab, Departamento de Física, Universidade Federal do Piauí, Teresina 64049-550, PI, Brazil; fwslima@gmail.com; 3Departamento de Física, Centro de Ciências Exatas e da Natureza CCEN, Universidade Federal da Paraíba, Cidade Universitária, João Pessoa 58051-970, PB, Brazil; 4Departamento de Física, Universidade Federal de Minas Gerais, C. P. 702, Belo Horizonte 30123-970, MG, Brazil; 5Department of Physics and Astronomy, University of Georgia, Athens, GA 30602, USA

**Keywords:** quantum spin models, phase diagram, exact diagonalization, ladder structure

## Abstract

The spin-1/2 quantum transverse Ising model, defined on a ladder structure, with nearest-neighbor and four-spin interaction on a plaquette, was studied by using exact diagonalization on finite ladders together with finite-size-scaling procedures. The quantum phase transition between the ferromagnetic and paramagnetic phases has then been obtained by extrapolating the data to the thermodynamic limit. The critical transverse field decreases as the antiferromagnetic four-spin interaction increases and reaches a multicritical point. However, the exact diagonalization approach was not able to capture the essence of the dimer phase beyond the multicritical transition.

## 1. Introduction

Low-dimensional quantum spin models have long been the subject of many theoretical and experimental approaches. The interest in such systems mainly derives from two factors. The first is the high experimental appeal because some physical realizations, when viewed by the relevant microscopic quantum interactions, consist, in fact, of single molecules, linear chains, or even ladder-like structures. To cite a few examples among the vast list of experimental results that fall into these scenarios, we have (i) molecular magnetism in Cu5-NIPA [1], V6-like magnetic molecules [2], and multiferroics [3] (these compounds can be considered zero-dimensional systems); (ii) TMMC, CsNiF${}_{3}$, and CuCl${}_{2}$2NC${}_{5}$H${}_{5}$ systems exhibiting one-dimensional characteristics [4]; (iii) ladder-type structures as in (VO)${}_{2}$P${}_{2}$O${}_{7}$ [5], Cu(C${}_{5}$H${}_{12}$N${}_{2}$)${}_{2}$2Cl${}_{4}$ [6], as well as La${}_{2}$CuO${}_{4}$ and La${}_{6}$Ca${}_{8}$Cu${}_{24}$O${}_{41}$ cuprates [7,8,9,10]. Of course, the majority of physical realizations indeed have the main quantum interactions along the three crystalline directions. Second, it is a real challenge to theoretically treat quantum models even in low spatial dimensions, since the statistical mechanics of a *d*-dimensional quantum model (*d* being the dimension of the lattice) is equivalent to a d+1-classical model [11], implying that even one-dimensional quantum systems can undergo a critical phase transition at zero temperature, the so-called *quantum phase transition*.

Regarding the low-dimensional theoretical models, the Ising, XY, and Heisenberg quantum spin systems have been widely studied in the literature, mostly with nearest-neighbor interactions and also in the presence of transverse fields (see, for example, Refs. [11,12]). In particular, spin-1/2 XY chains, with the transverse Ising case being a projection of the XY Hamiltonian when no interaction in the *y*-direction is present, have been studied not only for their static phase transition behavior but also from a dynamical perspective by computing the time-dependent correlation functions and the corresponding spectral functions [13,14,15,16]. On the other hand, spin-1/2 anisotropic Heisenberg chains have been treated according to different approaches, including the Bethe ansatz and Jordan–Wigner transformations (for example, see Ref. [17] and references therein). Furthermore, the effects of randomness in the isotropic XY chain [17], which makes the theoretical treatment even more difficult to tackle, have been introduced in the nearest-neighbor exchange and other types of interactions, as well as in the transverse field.

Despite the nearest-neighbor exchange being the relevant interaction in all physical realizations and models cited above, four-spin interactions in a plaquette have been suggested to reproduce the dispersion relation observed in inelastic neutron scattering experiments on cuprates [7,8,9,10] and even to stabilize a chiral spin liquid on a triangular lattice [18]. This new experimental motivation has lead to an increase in theoretical investigations of quantum Ising–Heisenberg-like models defined on a ladder [19,20], especially with four-spin interaction, which has been shown to induce other unusual types of order, such as scalar chirality and intra-rail staggered dimerization (see, for instance, Ref. [21] and references therein).

The spin-1/2 quantum transverse Ising model, which is the simplest model defined on a ladder structure with nearest-neighbor interaction, has thus been widely studied in the literature. In particular, the model including a four-spin interaction on a plaquette has been recently considered by using the density matrix renormalization group (DMRG) approach [21]. (The dynamical behavior has also been recently studied via the recurrence relation method [22]) The ground state phase diagram has been obtained in the transverse field versus four-spin interaction plane. In this case, a ferromagnetic-to-paramagnetic second-order phase transition is observed, and, for sufficiently high antiferromagnetic four-spin couplings, the system presents a dimer phase and a multicritical point. Although the transition for ferromagnetic four-spin couplings is well established, with the two degenerated ferromagnetic phases becoming equal to the paramagnetic phase at the transition point, the situation is less clear for antiferromagnetic four-spin couplings, where the ordered phase is given by a superposition of dimerized states [21]. In view of this fact, it is interesting to investigate this model with different techniques in order to obtain a better picture of the phases along the dimerized transition line. Since the finite-size-scaling approach, based on exact results on finite lattices, has been shown to be one of the most accurate methods to treat such models, it was employed here to obtain the quantum phase transition of the transverse Ising ladder with four-spin interactions. It would also be quite interesting and illustrative to compare the present results of the phase diagram to those obtained for the one-dimensional model with linear four-spin coupling, which have been obtained by using the same exact diagonalization procedure [23].

Thus, based on the above, in the present work, we have revisited the transverse Ising model with four-spin interaction in a ladder structure and used exact diagonalization on finite lattices. The energy gap has been computed, and, from the corresponding finite-size-scaling (FSS) relation, the critical transverse field has been obtained for several values of four-spin interaction. The ferromagnetic–paramagnetic phase transition line has been obtained by extrapolating the data to the thermodynamic limit. The results agree well with those previously obtained from the DMRG [21]. However, contrary to the model with four-spin interactions in one dimension [23], it has been noticed that, in the ladder structure, there are much stronger finite-size effects, and one indeed has to use the larger possible lattices for the extrapolations. In addition, the ferromagnetic–paramagnetic transition line on the ladder does not go to zero at the multiphase point in the classical limit of vanishing transverse field, as does the one-dimensional model.

The remainder of the paper is structured as follows. In the next section, the model is defined, and the configurations in the classical limit are discussed. Section 3 describes the theoretical approach using the exact diagonalization on finite ladders and the corresponding FSS relation for the energy gap and critical transverse field. The results of the transition line are presented in Section 4, and some concluding remarks are addressed in the final section.

## 2. Model

The Hamiltonian of the spin-1/2 Ising model, defined on a ladder structure, as depicted in Figure 1, with four-spin interaction and in the presence of a transverse field, can be written as
(1)$$H=-J\sum_{i=1}^{L}\left(\sigma_{1,i}^{z}\sigma_{1,i+1}^{z}+\sigma_{2,i}^{z}\sigma_{2,i+1}^{z}\right)-J\sum_{i=1}^{L}\sigma_{1,i}^{z}\sigma_{2,i}^{z}+J_{4}\sum_{i=1}^{L}\sigma_{1,i}^{z}\sigma_{1,i+1}^{z}\sigma_{2,i}^{z}\sigma_{2,i+1}^{z}-B\sum_{i=1}^{L}\left(\sigma_{1,i}^{x}+\sigma_{2,i}^{x}\right),$$
where J>0 is the nearest-neighbor exchange interaction along the rails and in the rungs, $J_{4}$ is the four-spin interaction connecting the spins in a plaquette, and *B* is the transverse external field applied in the *x* direction. The sums are over the *L* rungs of a ladder with periodic boundary conditions in the direction of the two side rails denoted, respectively, by 1 and 2. The spin-1/2 operators $\sigma_{\ell ,i}^{\alpha }$, with ℓ=1,2 and α=x,z, are given by the Pauli spin matrices.

Due to the positiveness convention of the four-spin interaction in the Hamiltonian (Equation 1), when $J_{4}<0$, the plaquette ordering is also ferromagnetic for low values of transverse fields. As a result, the system undergoes a second-order transition from a ferromagnetic phase to a paramagnetic phase at a critical transverse field value $B_{c}$. This critical transverse field decreases as $J_{4}$ increases. As has been shown in Ref. [21], when $J_{4}>0$, this transition persists until some value of $J_{4}$ (that is *B*-dependent) where a dimer phase is set up in the ladder.

At B=0, one has a classical system with a multiphase point at $J_{4}=3/2$ dividing the classical axis $J_{4}$ into two regions: (1) for $J_{4}<3/2$ a double degenerated ferromagnetic phase and (2) for $J_{4}\geq 3/2$ a ground state that is $2^{L+1}$ degenerate, with a residual entropy $S_{r}=\left(L+1\right)k_{B}\ln2$ (here, $k_{B}$ is the Boltzmann constant).

Contrary to the one-dimensional model with four-spin interaction along a straight line [23], the ferromagnetic–paramagnetic transition line does not terminate at the multiphase point at B=0. This is a consequence of the ladder structure allowing the appearance of the rung dimerized phase [21].

## 3. Theoretical Background

We have used finite ladders of length *L*, with N=2L sites, and periodic boundary conditions. For each ladder, the energy gap $G_{L}\left(B,J_{4}\right)$ has been computed. The quantity $G_{L}\left(B,J_{4}\right)$ is given by
(2)$$G_{L}\left(B,J_{4}\right)=E_{L}^{1}-E_{L}^{0},$$
where $E_{L}^{1}-E_{L}^{0}$ is the energy gap between the first excited state and the ground state, respectively.

$G_{L}\left(B,J_{4}\right)$ is equivalent to the correlation length in thermal systems and satisfies the FSS relation [24]
(3)$$LG_{L}\left(B_{c}^{L},J_{4}\right)=L^{\prime }G_{L^{\prime }}\left(B_{c}^{L},J_{4}\right),$$
for two finite ladders of length *L* and $L^{\prime }$, with usually $L>L^{\prime }$. From the above relation, it is possible to estimate $B_{c}^{L}$, the critical field for the ladder pair ($L,L^{\prime }$).

For sufficiently large lattices, it is expected that the quantities $LG_{L}\left(B,J_{4}\right)$, as a function of *B* for a given value of $J_{4}$, cross at the same point $B_{c}$, the critical transverse field for the considered value of $J_{4}$. However, in some cases, where the size of the finite lattices is insufficient, residual corrections make the crossing points $B_{c}^{L}$ suffer a systematic shift as *L* varies. Nevertheless, there is an additional FSS relation for the *L* dependence on the transition points $B_{c}^{L}$ given by [25]
(4)$$B_{c}^{L}=B_{c}+aL^{-1/\nu }\left(1+bL^{-\omega }\right),$$
where $B_{c}=B_{c}^{L\to \infty }$ is the critical transverse field in the thermodynamic limit, ν is the correlation length critical exponent, ω is the correction-to-scaling exponent, and *a* and *b* are non-universal constants. In this way, for every chosen value of $J_{4}$ and reference ladder $L^{\prime }$, one obtains the crossing point $B_{c}^{L}$ for the ladder pair ($L,L^{\prime }$) through Equation (Equation 3). Next, using Equation (Equation 4) for various values of *L*, the desired extrapolated value of critical transverse field can be computed.

In the present work, the energy gap has been obtained through exact diagonalization of finite ladders with length *L* in the range 2≤L≤11. For the larger lattices, we have employed the Lanczos diagonalization procedure [26].

## 4. Results and Discussion

In what follows, J=1. This value of *J* can also be interpreted as measuring the four-spin interaction $J_{4}$, as well as the transverse field strength *B*, in units of *J*.

As an example of the behavior of energy gap as a function of the transverse field, Figure 2 shows $LG_{L}\left(B,J_{4}\right)$ for several values of *L* and $J_{4}=0$. In this case, we simply have the transverse Ising model on a ladder [27]. It is clear that a small region of unique crossings is only achieved for the larger ladders. The estimate of the critical transverse field, in this case, is revealed by the corresponding arrow.

It is also evident in the figure that for smaller ladders, there is a shift in the values of crossings $B_{c}^{L}$. We can thus compute $B_{c}^{L}$ by taking a reference lattice $L^{\prime }$ and several values of $L>L^{\prime }$. From the results of Figure 2, it is possible to obtain a clear crossing by using $L^{\prime }=2,3$, and 4. The results obtained are depicted in Figure 3, which allows one to obtain additional extrapolated estimates of $B_{c}$ through fits to Equation (Equation 4) for each value of the reference ladder $L^{\prime }$. The present estimate $B_{c}=$1.8322(2) agrees very well with $B_{c}=1.83213$ from the DMRG of Ref. [21] using a finite ladder with length L=28.

It should be stressed that in using Equation (Equation 4), the exponents ν and θ are required. Although from Equation (Equation 3) it is possible to compute ν using renormalization group ideas and estimates of $B_{c}$ [23,28] (for instance, from the values of the crossings $B_{c}^{L}$ for the larger ladders), the obtained results are close to the exact ones for this universality class. However, the correction-to-scaling exponent cannot be obtained in this way and, at least, should be treated as an extra adjustable parameter. For this reason, and due to the fact that we are indeed more interested in the location of the transition line, we have resorted to the known values ν=1 and θ=2 for the Ising universality class in order to compute the extrapolated critical transverse field.

Similar curves, with similar FSS analysis, are obtained when $J_{4}>0$. Figure 4 shows the critical transverse field $B_{c}$ as a function of $J_{4}$ obtained from the present method in comparison to some results from the DMRG of Ref. [21]. One can clearly see that, contrary to the very same model in one dimension [23] (with four-spin interactions along a straight line), the transition line definitely does not go to zero as $J_{4}$ tends to 3/2, the multiphase point. The agreement with DMRG results is not only apparent in Figure 4 but also in Table 1, which gives a more detailed numerical comparison for some selected values of $J_{4}$.

## 5. Concluding Remarks

The transverse Ising model, defined on a ladder and with four-spin interactions on a plaquette, has been studied by using the finite-size-scaling approach with exact diagonalization of finite ladders. The method has achieved acceptable results for the ferromagnetic-to-paramagnetic transition line in the transverse field versus four-spin interaction plane. The computed critical transverse field in the thermodynamic limit is also in good agreement with that from the density matrix renormalization group procedure.

Despite the expected efficacy of the FSS along the ferromagnetic–paramagnetic transition line, the model shows, in fact, strong finite-size effects, and only with considerably larger lattices could the results be accurately obtained. The finite-size effects here are, in fact, stronger than those seen in the one-dimensional model [28]. The ferromagnetic–paramagnetic transition line definitely does not meet the multiphase point at zero field, and there is no reentrancy along this transition line whatsoever. This behavior is contrary to the one-dimensional model although in agreement with the DMRG approach providing a dimerized phase for sufficiently high values of antiferromagnetic four-spin interaction.

One unexpected feature is the exact diagonalization on finite lattices method being unable to locate the dimerized phase and the corresponding transition to the ferromagnetic and paramagnetic phases that occurs. For values $J_{4}>1.9$, the crossings become more diffuse, and in some regions it is not possible to obtain good fits with the expected scaling relation. More suitable quantities than the energy gap may be necessary to unveil the transition characteristics and the microscopic spin behavior for larger values of four-spin interaction by using the FSS procedure. Accordingly, a detailed characterization of the nature of the multicritical point, whether tricritical, tetracritical, etc. [29], remains to be conducted. 

## Figures and Tables

**Figure 1 entropy-25-01665-f001:**
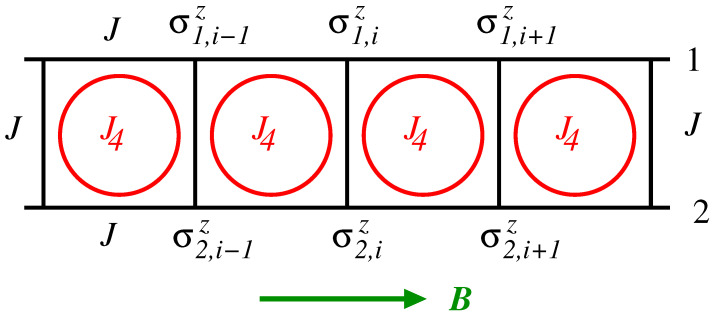
Sketch of a ladder with rails labeled as 1 and 2. The interaction along the rails and in the rungs is *J*. The circles represent the four-spin interaction $J_{4}$ in each plaquette. $\sigma_{n,i}^{z}$, with n=1,2, are the *z*-component spin operators. The *z* axis is taken to be perpendicular to the plane of the figure and the transverse field *B* is along the *x* axis.

**Figure 2 entropy-25-01665-f002:**
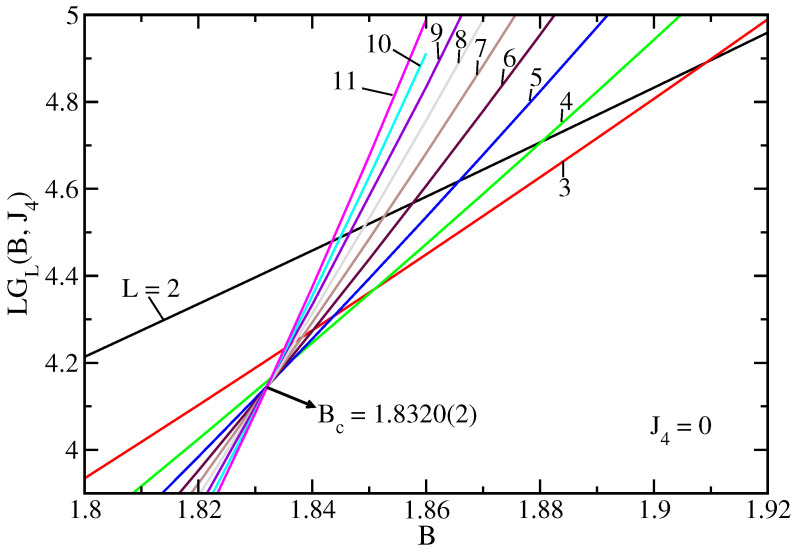
The quantity $LG_{L}\left(B,J_{4}\right)$, defined in the text, as a function of the transverse field *B*, for several values of *L* (indicated in the curves) and $J_{4}=0$. The arrow indicates the estimate of $B_{c}$ when considering only the larger ladders.

**Figure 3 entropy-25-01665-f003:**
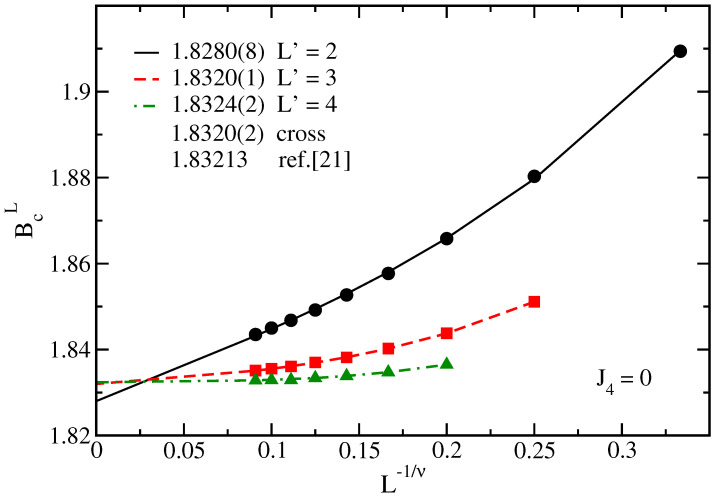
Crossing transverse field $B_{c}^{L}$, as a function of $L^{-1/\nu }$ (ν=1), for $J_{4}=0$ and different values of the reference ladder $L^{\prime }$ (indicated in the legend). In the legend we find the extrapolated values of $B_{c}$, using fits to Equation (Equation 4) with θ=2, for each curve. Furthermore, indicated in the legend come the values from the crossings of the larger ladders in Figure 2 (cross) and from Ref. [21].

**Figure 4 entropy-25-01665-f004:**
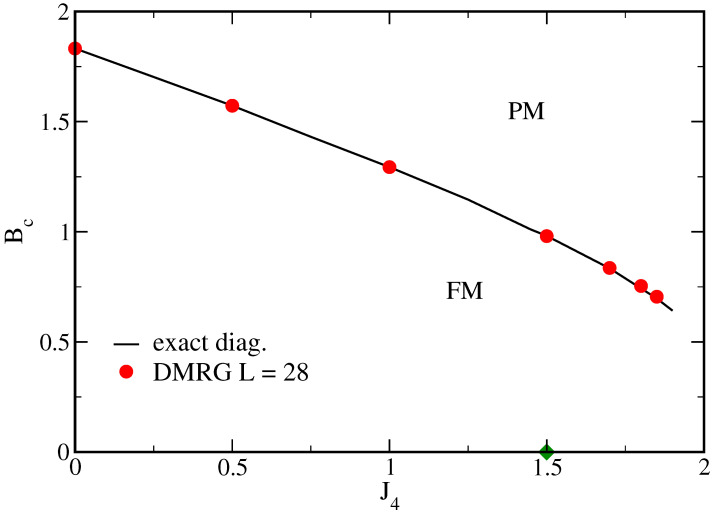
Critical values of the transverse field $B_{c}$ as a function of the four-spin interaction $J_{4}$. FM and PM stand for the ferromagnetic and paramagnetic phases, respectively. The full line represents the present results and the circles the values from the DMRG of Ref. [21]. The diamond is located at the multiphase point at $B_{c}=0$.

**Table 1 entropy-25-01665-t001:** Critical values of the transverse field $B_{c}$ in the thermodynamic limit for some values of the four-spin interaction $J_{4}$. The first row gives the results from the present work and the second row those from the DMRG calculation [21] with L=28.

$J_{4}$	0	0.5	1	1.5	1.7	1.85	
$B_{c}$	1.8322(2)	1.5726(2)	1.2938(2)	0.9813(3)	0.834(3)	0.697(2)	this work
	1.83213	1.57226	1.29334	0.98041	0.83579	0.70510	DMR [21]

## Data Availability

Data are contained within the article.

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
