# Peer review of "Ising Ladder with Four-Spin Plaquette Interaction in a Transverse Magnetic Field"

_entropy, 2023, doi:10.3390/e25121665_

Round 1

Reviewer 1 Report

Comments and Suggestions for Authors

Reviewer report on the manuscript entropy-2728045: "Ising ladder with four-spin plaquette interaction in a transverse  magnetic field" by M. E. S. Nunes, F. W. S. Lima, J. A. Plascak

The manuscript is devoted to a detailed analysis of a spin-1/2 Ising two-leg ladder with a four-spin plaquette interaction in a transverse magnetic field, which was investigated through the exact diagonalization supplemented with a finite-size-scaling method. Within this method, the magnetic-field-driven quantum phase transition between the ferromagnetic and paramagnetic phases was examined as a function of the four-spin plaquette interaction. The manuscript is clearly written, the results seem to be correct and the relevant discussion is without any faulty judgement. I have only a few minor comments on the manuscript.

1. The second sentence in the abstract: "The quantum phase transition between the ferromagnetic and paramagnetic phases has been obtained in the thermodynamic limit" is somewhat confusing, because the used exact diagonalization method was limited to a relatively small finite-size systems and the thermodynamic limit was only obtained by the extrapolation procedure. This fact should be clarified in this sentence.

2. The introduction of the paper is too brief and it does not provide a sufficient background including all relevant references. Namely, the present paper should be put into a more proper context when mentioning in the introduction a few closely related papers on the Ising, Ising-Heisenberg and Heisenberg two-leg ladders with/without four-spin interaction and/or transverse magnetic field.

3. As the extrapolated exact diagonalization data are generally in a perfect agreement with DMRG results, it would be quite beneficial to the readers if the authors could add a short comment on the critical line shown in Fig. 5 along with the multiphase quantum critical point at J_4=1.5. Is it expected that the critical line will bend towards the multiphase quantum critical point at J_4=1.5? If so, there appears another related question why the exact diagonalization and DMRG data do not capture reentrance at J_4 \gtrsim 1.5?

4. There are a few small misprints in the words, which could be easily removed by running spellcheck.

To conclude, the present manuscript meets all high standards of the journal Entropy and it deserves right to be published there after a few minor corrections mentioned in this report will be taken into account.   

Comments on the Quality of English Language

see point 4. of the reviewer report

Author Response

The manuscript is devoted to a detailed analysis of a spin-1/2 Ising two-leg ladder with a four-spin plaquette interaction in a transverse magnetic field, which was investigated through the exact diagonalization supplemented with a finite-size-scaling method. Within this method, the magnetic-field-driven quantum phase transition between the ferromagnetic and paramagnetic phases was examined as a function of the four-spin plaquette interaction. The manuscript is clearly written, the results seem to be correct and the relevant discussion is without any faulty judgement. I have only a few minor comments on the manuscript.

1. The second sentence in the abstract: "The quantum phase transition between the ferromagnetic and paramagnetic phases has been obtained in the thermodynamic limit" is somewhat confusing, because the used exact diagonalization method was limited to a relatively small finite-size systems and the thermodynamic limit was only obtained by the extrapolation procedure. This fact should be clarified in this sentence. 

This was indeed misleading in the text. We have clarified, in the Abstract, that the results have been achieved through an extrapolation of the data obtained in finite ladders.

2. The introduction of the paper is too brief and it does not provide a sufficient background including all relevant references. Namely, the present paper should be put into a more proper context when mentioning in the introduction a few closely related papers on the Ising, Ising-Heisenberg and Heisenberg two-leg ladders with/without four-spin interaction and/or transverse magnetic field.

Please, see the new second paragraph in the Introduction, where we have mentioned some related papers regarding Ising-Heisenberg-XY models. Some additional references have also been added in the third paragraph. New references are [12-17] and [19,20].

3. As the extrapolated exact diagonalization data are generally in a perfect agreement with DMRG results, it would be quite beneficial to the readers if the authors could add a short comment on the critical line shown in Fig. 5 along with the multiphase quantum critical point at $J_4=1.5$. Is it expected that the critical line will bend towards the multiphase quantum critical point at $J_4=1.5$? If so, there appears another related question why the exact diagonalization and DMRG data do not capture reentrance at $J_4 > 1.5$?

In fact, this line ends at the multicritical point for B≠0 and no reentrancy has been achieved using the present FSS. We have added a new comment on this point in the end of the second paragraph of the final section.

4. There are a few small misprints in the words, which could be easily removed by running spellcheck.

Thanks for pointing us out some misprints. Corrected.

To conclude, the present manuscript meets all high standards of the journal Entropy and it deserves right to be published there after a few minor corrections mentioned in this report will be taken into account.   

We thank the referee for a positive evaluation of our paper.

Reviewer 2 Report

Comments and Suggestions for Authors

I don't understand what the purpose of the paper is. Ref.13 (with overlapping authorship) has already reported on DMRG simulations of the same model. The present work does not improve on the previous results of Ref.13, reproducing a previous work with what is considered an inferior technique (ED vs DMRG), at smaller system sizes. The authors have not argued the merits of this work.

Author Response

I don't understand what the purpose of the paper is. Ref.13 (with overlapping authorship) has already reported on DMRG simulations of the same model. The present work does not improve on the previous results of Ref.13, reproducing a previous work with what is considered an inferior technique (ED vs DMRG), at smaller system sizes. The authors have not argued the merits of this work.

We believe indeed that we haven't made clear enough in the text the merit of the present work. In Ref. [13] the phase diagram of the model has been obtained as a function of the 4-spin interaction using DMRG. This phase diagram is completely different from the one-dimensional version as given in Ref. [28]. A dimerized phase has been detected and its transition line to the paramagnetic phase obtained. The ferromagnetic-paramagnetic line is well understood. The dimerized-paramagnetic line is not, in our point of view, so clear as the former one, due to the dimerized phase being given by a quantum superposition of dimer states. This makes unclear the nature of the multicritical point. Naturally, studies employing different techniques should be very welcome to corroborate or not this new behavior. Unfortunately, the exact diagonalization and FSS were unable to unveil this character. However, it should be interesting to have a knowledge that this powerful technique could not enhance the insight of this transition. 

We have then made a discussion on this point in the final part of the fourth paragraph in the Introduction as well as in the second paragraph and the last sentence of the third paragraph of the final section.

Reviewer 3 Report

Comments and Suggestions for Authors

This work studies a transverse Ising model on a ladder with the nearest neighbor and four-spin interactions, using the exact numerical diagonalization up to eleven spins and the finite-size scaling technique. The authors present accurate critical values of the transverse field as a function of the strength of the four-spin interaction, that separates the ferromagnetic and paramagnetic phases. The result is throughly consistent with the previous result obtained by the density matrix renormalization group method. 

The way of numerical study is described clearly and the results are sound. However, the present work does nothing more than reproduce the results given in ref. [13] (Xavier et al., Phys. Rev. B 105 024430 (2022)) by using another well-known method. Nothing new is available. Therefore the present manuscript does not deserve to be published in Entropy.

Comments on the Quality of English Language

The language is good.

Author Response

This work studies a transverse Ising model on a ladder with the nearest neighbor and four-spin interactions, using the exact numerical diagonalization up to eleven spins and the finite-size scaling technique. The authors present accurate critical values of the transverse field as a function of the strength of the four-spin interaction, that separates the ferromagnetic and paramagnetic phases. The result is throughly consistent with the previous result obtained by the density matrix renormalization group method. 

The way of numerical study is described clearly and the results are sound. However, the present work does nothing more than reproduce the results given in ref. [13] (Xavier et al., Phys. Rev. B 105 024430 (2022)) by using another well-known method. Nothing new is available. Therefore the present manuscript does not deserve to be published in Entropy.

We believe indeed that we haven't made clear enough in the text the merit of the present work. In Ref. [13] the phase diagram of the model has been obtained as a function of the 4-spin interaction using DMRG. This phase diagram is completely different from the one-dimensional version as given in Ref. [28]. A dimerized phase has been detected and its transition line to the paramagnetic phase obtained. The ferromagnetic-paramagnetic line is well understood. The dimerized-paramagnetic line is not, in our point of view, so clear as the former one, due to the dimerized phase being given by a quantum superposition of dimer states. This makes unclear the nature of the multicritical point. Naturally, studies employing different techniques should be very welcome to corroborate or not this new behavior. Unfortunately, the exact diagonalization and FSS were unable to unveil this character. However, it should be interesting to have a knowledge that this powerful technique could not enhance the insight of this transition. 

We have then made a discussion on this point in the final part of the fourth paragraph in the Introduction as well as in the second paragraph and the last sentence of the third paragraph of the final section.

Round 2

Reviewer 1 Report

Comments and Suggestions for Authors

The authors have satisfactorily revised their manuscript according to the suggestions from my previous reports, so I suggest to accept it for publication in Entropy journal. In the second paragraph of Introduction I have only noticed 2 small misprints in the new red text:

1. "no interaction in the y-direction is present" should be replaced with "no interaction in the z-direction is present"

2. "Jordan-Wiegner" should be replaced with "Jordan-Wigner"

With these two minor changes, the manuscript can be accepted in its present form.

Reviewer 3 Report

Comments and Suggestions for Authors

The manuscript has been revised so as to address the merit of this work. However, as the authors mentioned in the manuscript, the present work using exact diagonalization along with the finite-size scaling (FSS) could not reveal the transition which was anticipated for a strong four-spin interaction. The message of this manuscript is only that the exact diagonalization and FSS cannot clarify as much as DMRG. I do not think this result deserves to be published. I would like to encourage the authors to pursue the transition associated with the dimerized states.

Comments on the Quality of English Language

The language is good.